# Adiposity measures and arterial stiffness in primary care: the MARK prospective observational study

Leticia Gomez-Sanchez,[1,2] Luis Garcia-Ortiz,[1,2,3] Maria C Patino-Alonso,[1,2,4] Jose I Recio-Rodriguez,[1,2,5] Fernando Rigo,[6] Ruth Martí,[7,8] Cristina Agudo-Conde,[1,2] Emiliano Rodriguez-Sanchez,[1,2,9] Jose A Maderuelo-Fernandez,[1,2] Rafel Ramos,[7,8,10] Manuel A Gomez-Marcos,[1,2,9] for the MARK Group

► Prepublication history and additional material are available online. To view these files please visit the journal online (http://dx.doi.org/10.1136/bmjopen-2017-016422).

For numbered affiliations see end of article.

**Correspondence to**
Leticia Gomez-Sanchez;
Leticiagmzsnchz@gmail.com

## ABSTRACT

**Background** The cardiovascular risk of obesity is potentially increased by arterial stiffness.

**Objective** To assess the relationship of adiposity measures with arterial stiffness in Caucasian adults with intermediate cardiovascular risk.

**Setting** Six Spanish health centres.

**Participants** We enrolled 2354 adults (age range, 35–74 years; mean age, 61.4±7.7 years, 61.9% male).

**Methods** This is a cross-sectional study that analyses data from the baseline visit of the improving interMediAte RisK management (MARK) study. The main outcome variables were body mass index (BMI), waist-to-height ratio (WHtR), Clínica Universidad de Navarra-body adiposity estimation (CUN-BAE) body fat percentage and body roundness index (BRI). Vascular function was assessed by the cardio-ankle vascular index (CAVI) with the VaSera device; brachial-ankle pulse wave velocity (baPWV) was determined using a validated equation.

**Results** The mean adiposity measures were a BMI of 29.2±4.4, WHtR of 0.61±0.07, CUN-BAE of 35.7±1.7 and BRI of 5.8±1.7. The mean stiffness measures were a CAVI of 8.8±1.2 and baPWV of 14.9±2.5. In multiple linear regression analyses, all adiposity measures were negatively associated with CAVI and baPWV (p<0.01 for all) after adjustment for possible factors of confusion. The proportion of CAVI variability via the adiposity measures were 5.5% for BMI, 5.8% for CUN-BAE, 3.8% for WHtR and 3.7% for BRI. These were higher among diabetic, obese, younger (≤62 years) and non-hypertensive subjects who had similar activity and sedentary profiles.

**Conclusions** Adiposity measures are negatively associated with arterial stiffness measures. The percentage of variation in CAVI explained by its relation to the different measures of adiposity ranges from 5.8% (CUN-BAE) to 3.7% (BRI). In the case of baPWV, it oscillates between 0.7% (CUN-BAE and BMI) and 0.1% (WHtR).

**Trial registration number** NCT01428934.

## INTRODUCTION

Obesity causes increased morbidity and mortality both globally and via cardiovascular

### Strengths and limitations of this study

► This is the first study to investigate the association between adiposity measures using a cardio-ankle vascular index (CAVI) with brachial-ankle pulse wave velocity (baPWV) in Caucasian adults with intermediate cardiovascular risk.

► All adiposity measures were negatively associated with CAVI and baPWV.

► The arterial stiffness variability was better explained for body mass index and Clínica Universidad de Navarra-body adiposity estimation as well as when CAVI was used as a measure of stiffness rather than baPWV.

► The most important limitation of this study is its transversal design, which prevents the establishment of causal relationships. It is also limited by the direction of influence of adiposity measures on arterial stiffness.

diseases.[1] However, the pathophysiological mechanisms that explain how obesity increases cardiovascular diseases complementary to classic risk factors are not well clarified.[2] It has been suggested that increased arterial stiffness is the pathological pathway through which obesity increases cardiovascular disease regardless of other classical risk factors.[3] We now know that the arterial stiffness measured with the brachial-ankle pulse wave velocity (baPWV) has independent capacity to predict coronary arteriosclerosis and mortality in the general population[4] and in subjects with diabetes.[5 6] Similarly, the cardio-ankle vascular index (CAVI) is related to arteriosclerotic disease in carotid and coronary arteries[7–9] and can predict new cardiovascular events in obese subjects.[10] However, the association between adiposity and arterial stiffness remains controversial. In this respect, there are studies that show that the body mass index (BMI) has an independent relationship

with arterial stiffness in the general population[3 11] and in patients with diabetes.[12 13] However, other research has not yet found this association,[14] or the association disappeared after adjusting for potential confounders,[13] or it showed a negative association.[15 16]

Other studies suggested a stronger correlation of measures in central or visceral adiposity than measures of general adiposity with arterial stiffness in the general population[3 17–20] in patients with diabetes[13] and patients with hypertension.[21] Finally, the Whitehall II Cohort study[22] showed that all measures of general adiposity, central adiposity and body fat percentage (BF%) were predictors of accelerated arterial stiffness in adults. In this context, the analysis of the relationship between arterial stiffness and different measures of adiposity can explain the role of obesity in cardiovascular disease.

Our study was designed bearing in mind that cardiovascular events occur more frequently in subjects with intermediate cardiovascular risk[23] and that there is are a lack of studies analysing the relationship of different adiposity measures with arterial stiffness in these subjects. The principal aim of this study was to analyse the relationship between adiposity measures and arterial stiffness in Caucasian adults with intermediate cardiovascular risk. The secondary aim was to investigate the changes between the associations of adiposity measures with distinct arterial stiffness markers.

## METHODS
### Study design
The results presented here correspond to a subanalysis of the MARK study to data collected at the baseline visit. The characteristics and form of selection of the subjects—as well as the methodology of the measurements made in the subjects included in the *improving interMediAte RisK management (MARK)* study (NCT01428934)—has been published in the same protocol[24] and in previous publications from the MARK study group.[25 26] The MARK study was a cross-sectional study whose main objective was to evaluate whether the ankle-brachial index (ABI), the CAVI, postprandial glucose, glycosylated haemoglobin, pressure arterial self-measurement and the presence of comorbidities are independently associated with the incidence of vascular events and whether these measures can improve the predictive capacity of the current risk equations. The second part is a follow-up of 5 years and 10 years to estimate cardiovascular morbidity and mortality.[24]

### Study population
The population was enrolled at six primary care centres from different regions of Spain. The data collection was from July 2011 to June 2013. The study included subjects aged 35 and 74 years who had intermediate cardiovascular risk. It was defined as coronary risk between 5% and 15% at 10 years according to the Framingham-adapted risk equation[27]: cardiovascular mortality risk between 1%

**Figure 1** Flow chart. MARK substudy. ABI, ankle-brachial index; baPWV, brachial-ankle pulse wave velocity; CAVI, cardio-ankle vascular index; WC, waist circumference.

and 5% at 10 years according to the SCORE equation[28] or a moderate risk according to the European Society of Hypertension guidelines for the management of arterial hypertension.[29]

The exclusion criteria were terminal illness, institutionalisation at the appointment time or a personal history of atherosclerotic disease registered in his/her electronic clinical history. This study analysed 2354 of the 2495 subjects recruited in the MARK study. In this analysis, we excluded 141 individuals with ABI ≤0.9 (n=99) or whose CAVI (n=16), baPWV (n=12) and WC (n=14) measurements were incomplete (figure 1).

### Anthropometric measurements
Weight was achieved using a certified electronic scale (Seca 770, medical scale and measurement systems, Birmingham, UK) after adequate calibration (precision ±0.1 kg). The subjects' weights were rounded to the nearest 100 g. Height was achieved using a stadiometer (Seca 222). We used the average of two measurements of body weight and height. To calculate the BMI, we used the following formula: BMI=weight (kg)/height squared (m$^2$). The waist circumference was measured according to the 2007 recommendations of the Spanish Society for the Study of Obesity.[30] The measurements were executed with the subjects standing, wearing no shoes and in light clothing. To calculate waist-to-height ratio (WHtR), we use the following formula: WHtR=waist circumference (cm)/height (cm).[31 32]

The BF% was calculated according to the Clínica Universidad de Navarra-body adiposity estimator (CUN-BAE) using the recommendations of Gómez-Ambrosi *et al*[33]: BF%=−44.988+ (0.503 × age) + (10.689 × sex) + (3.172 × BMI) − (0.026 × BMI$^2$ + (0.181 × BMI × sex) − (0.02 BMI × age) − (0.005 × BMI$^2$ × sex) + (0.00021×BMI$^2$ × age). Here, male=0 and female=1 for sex.

The body roundness index (BRI) was calculated using the following formula: BRI=364.2−365.5 × $\sqrt{1 - \left( \frac{(WC/(2\pi))^2}{(0.5\ height)^2} \right)}$.[34] The BRI can estimate the percentage of body fat and visceral adipose tissue.

## CAVI and baPWV

The device VaSera VS-1500 device (Fukuda Denshi) was used to measure the CAVI.[35 36] The CAVI values were calculated by estimating the stiffness parameter β using the following equation: $\beta = 2\rho \times 1/(Ps − Pd) \times \ln(Ps/Pd) \times PWV^2$, where ρ is blood density, Ps and Pd are SBP and DBP in mm Hg and PWV was measured between the aortic valve and the ankle.[37] The mean coefficient of variation of CAVI values was less than 5% indicating that it is a reproducible measure facilitating clinical use.[36]

The baPWV was estimated using the equation baPWV=(0.5934 × height (cm) + 14.4724)/tba (where tba is the time interval between the arm and ankle waves).[38]

One hour before the measurements, subjects were fasted from smoking and physical exercise. The measurements were performed after 10 min in supine decubitus and with stable temperatures.

## Definition of hypertension, type 2 diabetes mellitus and dyslipidaemia

The subjects were diagnosed with hypertension following the criteria of the European Society of Hypertension and European Society of Cardiology (ESC) guidelines (values ≥140 mm Hg systolic blood pressure (SBP) and/or ≥90 mm Hg diastolic blood pressure (DBP) or the presence of antihypertensive treatment)[29] or if they were previously diagnosed with hypertension. Type 2 diabetes mellitus was diagnosed following the criteria of the American Diabetes Association: presence of HbA1c ≥6.5% or fasting plasma glucose ≥126 mg/dL or 2-hour plasma glucose ≥200 mg/dL during an oral glucose tolerance test or in a patient with classic symptoms of hyperglycaemia or hyperglycaemic crisis, a random plasma glucose ≥200 mg/dL or the presence of antidiabetic treatment.[39] Subjects were also included if they had been previously diagnosed with type 2 diabetes mellitus. Dyslipidaemia was defined if they were treated with lipid-lowering drugs or had altered low-density lipoprotein (LDL) ≥130 mg/dL, high-density lipoprotein (HDL) ≤45 mg/dL in men and ≤55 in women and triglyceride (TG) ≥150 mg/dL as established by the ESC and the European Atherosclerosis Society 2011.[40]

## Office or clinical blood pressure

We performed three measurements of SBP and DBP using a validated OMRON model M10-IT sphygmomanometer (Omron Health Care, Kyoto, Japan). The blood pressure was taken following the recommendations of the European Society of Hypertension.[41] The measure was the average of the last two measurements.

## Lifestyles
### Tobacco

Data were collected whether the subjects smoked or not. We defined smokers as those who currently smoke or who have not smoked for a year.[24]

### Leisure time physical activity (LTPA)

LTPA was measured with the Minnesota LTPA Questionnaire[42] validated in Spanish population in males and females.[43 44] The questionnaire was collected by previously trained interviewers, and the time spent in the registry was between 10 and 20 min per subject. Information was collected about the type, duration and frequency of physical activity (PA) for each subject during the previous year. Each PA was assigned a code according to its intensity. This code is based on the quotient obtained by dividing the metabolic rate during the practice of BP between the basal metabolic rate (MET).[45] We assume that 1 MET equals 1 kcal/min of energy expenditure. Thus, we calculated the total energy expenditure during the free time of PA ($EEPA_{total}$) in kilocalories per week. By considering the intensity code of the PA, we could measure the energy expenditure in PA (EEPA) according to the classification of the PA performed (intense, moderate or light) as follows: we considered light intensity PA at <4 MET such as walking ($EEPA_{light}$). Moderate intensity PA was 4–5.5 METs such as brisk walking ($EEPA_{moderate}$). Intense intensity PA was greater than or equal to 6 METs such as jogging ($EEPA_{intense}$). Therefore, in each individual, the: $EEPA_{total}=EEPA_{light} + EEPA_{moderate} + EEPA_{intense}$.[46]

Following the recommendations of the American Heart Association,[47] we considered the subjects sedentary when they did not comply with the recommendations of practicing moderate intensity PA for a minimum of 30 min 5 days per week ($EEPA_{moderate}$ <675 kcal/week) or high-intensity aerobic PA practice for a minimum of 20 min on 3 days each week ($EEPA_{intense}$ <420 kcal/week).[46]

### Laboratory determinations

After 12 hours of fasting from eating and smoking, a blood test was performed at the health centre between 08:00 and 09:00 hours. Plasma glucose, HDL cholesterol concentrations and TG concentrations were measured using standard enzymatic automated methods. The Friedewald equation was used to calculate LDL cholesterol. The atherogenic index was calculated as total cholesterol/HDL cholesterol. The researchers who performed the assays different tests were blinded to the clinical data of the participants. The different tests were carried out within a maximum period of 10 days.

### Data analysis

Descriptive statistics were expressed as the mean±SD for continuous variables or number (%) for categorical variables. The $\chi^2$ test or the Fisher's exact test analysed the association between independent categorical variables. Quantitative variables were compared using Student's t-test. Pearson's correlation coefficient was used

to estimate the relationship of the adiposity measures to CAVI and baPWV. We used Steiger's Z statistics to test the significance of the difference between correlation coefficients.[48]

Four different multiple linear regression models were used to study the associations of each adiposity measure with CAVI, and four other models were used with baPWV. The CAVI and baPWV were the dependent variables, and the adiposity measures were the independent variables in each model. All models were adjusted for age (years), sex (0=male and 1 = female), SBP, smoking status (0=no and 1=yes), atherogenic index, HbA1c and METs/min/week. The explanatory capacity of the model was measured by $R^2$, and the proportion attributed to each variable was estimated by the change in $R^2$. The analysis was also performed via age groups, diagnosis of diabetes, hypertension and obesity.

Analysis of covariance models were used to test the differences in the mean values of CAVI and baPWV with the quartiles of the four adiposity measures after adjusting for the confounding variables that were used in regression analysis. Pairwise post hoc comparisons were studied using the Bonferroni test. Data were evaluated using SPSS Statistics for Windows V.23.0. We defined significant differences at p<0.05.

### Ethics statement

Before inclusion, all participants were informed about the objectives, tests to be performed and the need to sign the consent to participate in the study. The study was approved by the independent ethics committee of the Primary Care Research Institute Jordi Gol, the Health Care Area of Salamanca and Palma of Mallorca. The study followed the recommendations of the Declaration of Helsinki.[49] The confidentiality of the information provided by participants was ensured by complying with the rules established by Spanish Organic Law 15/1999 of 13 December on the Protection of Personal Data.

### RESULTS

Anthropometric measures, clinical characteristics and vascular function measures of the subjects are presented in table 1. The mean age of the patients was 61.4±7.7 years, and 61.9% were male. Male subjects constituted a higher percentage of smokers (31.5 vs 22.7) and hypertension (80.1 vs 75.4) compared with females. However, females had a higher prevalence of obesity (40.4 vs 33.4), sedentariness (53.7 vs 37.0), dyslipidaemia (73.1 vs 63.6) and diabetes (36.5 vs 31.8) relative to men. The mean value of CAVI was 8.8±1.2 (8.9 in males and 8.6 in females, p<0.001). The mean baPWV was 14.9±2.5 (14.8 in males and 15.0 in females). All of the adiposity measures except for waist circumference were higher in women than men.

Pearson's correlation coefficient results between the adiposity measures and the vascular function parameters are shown in table 2. All adiposity measures were negatively correlated with CAVI, and this correlation increases after adjusting for age, sex and SBP. The correlation between CAVI and baPWV was r=0.745 (p<0.001). We found differences in correlation coefficients between CAVI, baPWV and measures of adiposity (p<0.001 in all cases).

The online supplementary figure shows that the estimated marginal means of CAVI (A) and baPWV (B) by quartiles of the different adiposity measures. After adjustment for the variables used in the multiple linear regression analysis, the mean CAVI values decreased as the quartiles of the four adiposity measurements increased (p<0.05). However, the same is not true of baPWV with WHtR and BRI (p>0.05).

In the multiple linear regression analysis, CAVI and baPWV showed negative associations with all adiposity measures (p<0.01 for all) after adjustment for age, sex, SBP, smoking, atherogenic index, HbA1c and METs/min/week (table 3). The proportion of CAVI variability that can be attributed to the variation in the adiposity measures was 5.5% for BMI, 5.8% for CUN-BAE, 3.8% for WHtR and 3.7% for BRI. For baPWV, the variability by the measures of adiposity were 0.7% for BMI and CUN-BAE, 0.1% for WHtR and 0.2 for BRI. The association between adiposity measurements and CAVI revealed a standardised β between −0.450 (CUN-BAE) and −0.221 (WHtR). In the case of baPWV, the values oscillate between −0.152 (CUN-BAE) and −0.044 (WHtR).

In the multiple linear regression analysis by subgroup, the proportion of CAVI variability by adiposity measures was higher among diabetics, the obese, non-hypertensive, and subjects 62 years of age or younger; it was similar in active and sedentary people (table 4).

### DISCUSSION

This study showed that adiposity measures have a negative association with arterial stiffness, especially CAVI. BMI and CUN-BAE have the highest coefficient of determination. We found a negative association of different adiposity measures with CAVI and baPWV after adjustment for other variables of confusion.

In this study, the mean value of CAVI was higher in males, which concurs with published data indicating that CAVI increases linearly with age. The values of CAVI are higher in men than in women (approximately 0.2, which is equivalent to 4–5 years old).[22 50] We found no differences in the mean values of baPWV between sexes, which is consistent with data published by Tomiyama et al,[51] who showed that the effect of age on baPWV is different according to sex. Females have a higher arterial stiffness than prepubertal males—this increases after menopause. Men, however, experience a linear increase in arterial stiffness from puberty. This suggests that the large arteries of females are intrinsically more rigid than those in men. However, in women in reproductive age, the effects are offset by sex hormones.[52 53]

This negative association with CAVI has already been described in previous studies. BMI shows a negative

**Table 1** General characteristics of all the sample and by gender

| Variables | Global (n=2354) | Males (n=1456) | Females (n=898) | p Value |
|---|---|---|---|---|
| Age (years) | 61.4±7.7 | 61.1±8.1 | 61.8±7.0 | 0.030 |
| Smoking n (%) | 658 (28.0) | 456 (31.5) | 202 (22.7) | <0.001 |
| Alcohol (gr/week) | 72.2±117.5 | 102.2±133.3 | 23.6±59.9 | <0.001 |
| Physical activity (METs-min/week) | 2481±2512 | 2886±1831 | 1825±1691 | <0.001 |
| Sedentary n (%) | 1020 (43.3) | 538 (37.0) | 482 (53.7) | <0.001 |
| Height (cm) | 165±9 | 170±7 | 156±6 | <0.001 |
| Weight (kg) | 79.4±14.6 | 83.9±13.4 | 72.2±13.3 | <0.001 |
| BMI (kg/m$^2$) | 29.2±4.4 | 29.1±3.9 | 29.5±5.1 | 0.035 |
| BMI ≥30 n (%) | 847 (36.0) | 485 (33.4) | 362 (40.4) | 0.001 |
| Waist circumference (cm) | 100.9±11.6 | 102.9±10.5 | 97.6±12.5 | <0.001 |
| WHtR | 0.61±0.07 | 0.61±0.06 | 0.62±0.08 | <0.001 |
| CUN-BAE | 35.7±1.7 | 31.1±4.5 | 43.1±5.1 | <0.001 |
| BRI | 5.8±1.7 | 5.7±1.5 | 6.1±2.1 | <0.001 |
| SBP (mm Hg) | 137.1±17.4 | 138.9±17.1 | 134.2±17.5 | <0.001 |
| DBP (mm Hg) | 84.4±10.2 | 85.5±10.4 | 82.7±9.7 | <0.001 |
| Heart rate (beats per minute) | 74.2±10.2 | 73.3±12.7 | 75.8±11.6 | <0.001 |
| Hypertension n (%) | 1712 (72.7) | 1122 (80.1) | 590 (75.4) | <0.001 |
| Antihypertensive drugs (n (%)) | 1199 (50.9) | 729 (50.2) | 470 (52.6) | 0.289 |
| Total cholesterol (mg/dL) | 225.8±40.9 | 220.8±39.1 | 233.9±42.5 | <0.001 |
| LDL cholesterol (mg/dL) | 140.4±34.9 | 138.9±34.2 | 142.8±35.8 | 0.011 |
| HDL cholesterol (mg/dL) | 49.8±12.9 | 47.9±11.9 | 52.9±13.8 | <0.001 |
| Triglycerides (mg/dL) | 145.5±96.6 | 150.3±106.3 | 137.7±77.9 | 0.001 |
| Atherogenic index | 4.8±1.3 | 4.8±1.3 | 4.7±1.3 | 0.002 |
| Dyslipidaemia n ((%)) | 2151 (91.4) | 1311 (90.0) | 840 (93.5) | <0.001 |
| Lipid lowering drugs (n (%)) | 671 (28.5) | 392 (26.8) | 279 (31.0) | 0.034 |
| FPG (mg/dL) | 107.2±34.8 | 106.9±33.9 | 107.6±36.1 | 0.659 |
| HbA1c | 4.8±1.3 | 5.9±1.4 | 6.1±1.4 | 0.001 |
| Diabetes (n (%)) | 791 (33.6) | 463 (31.8) | 328 (36.5) | 0.020 |
| Antidiabetic drugs (n (%)) | 474 (20.1) | 269 (18.5) | 205 (22.9) | 0.011 |
| CAVI | 8.8±1.2 | 8.9±1.2 | 8.6±1.1 | <0.001 |
| baPWV (m/s) | 14.9±2.5 | 14.8±2.5 | 15.0±2.6 | 0.107 |

Values are means and (SD) for continuous data and number and (proportions) for categorical data. p Value differences between male and females. baPWV, brachial-ankle pulse wave velocity; BMI, body mass index; BRI, body roundness index; CAVI, cardio-ankle vascular index; CUN-BAE, Clínica Universidad de Navarra-body adiposity estimator; DBP, diastolic blood pressure; FPG, fasting plasma glucose; HbA1c, glycosylated haemoglobin; HDL, high-density lipoprotein; LDL, low-density lipoprotein; METs-min/week, metabolic equivalent minutes per week; SBP, systolic blood pressure; WHtR, waist-to-height ratio.

association in children,[54] and in patients with hypertension and in subjects with diabetes in Ghana.[55] Similarly, the waist circumference has a negative relationship in subjects with metabolic syndrome.[26 56]

However, other authors have described a positive association of different adiposity measures with the β-stiffness parameter, but after adjustment for age and other possible confounding factors, the association remained for only men with type 2 diabetes mellitus.[57] Other studies showed no association between BMI and CAVI.[58]

Studies analysing the association of adiposity measures with baPWV have also been performed mainly in Eastern populations and have assessed the association between BMI and waist circumference as measures of adiposity. The results are controversial with some finding a negative association with BMI[51] and with waist circumference in men or only in women.[59 60] However, other studies showed a positive correlation with BMI and waist circumference.[61] One study analysed the association of different adiposity parameters with baPWV in middle-aged adults and found a positive association with waist circumference and visceral fat but not with BF%.[62]

The results are also not consistent in studies that used the carotid-femoral pulse wave velocity (cfPWV)

**Table 2** Bivariate correlations of adiposity measures with CAVI and baPWV

| | CAVI | | baPWV | |
|---|---|---|---|---|
| | Unadjusted | Adjusted† | Unadjusted | Adjusted† |
| BMI | −0.264** | −0.303** | −0.035 | −0.068** |
| WHtR | −0.119** | −0.222** | 0.090** | 0.001 |
| CUN-BAE | −0.187** | −0.297** | 0.054* | −0.063** |
| BRI | −0.125** | −0.218** | 0.078** | 0.005 |

The correlation coefficients between CAVI, baPWV and adiposity measurements showed significant differences (p<0.001 in all cases).p Values by Pearson correlation: *p<0.05, **p<0.01.†Adjusted for age, sex and systolic blood pressure. baPWV, brachial-ankle pulse wave velocity; BMI, body mass index; BRI, body roundness index; CAVI, cardio-ankle vascular index; CUN-BAE, Clínica Universidad de Navarra-body adiposity estimator; WHtR, waist-to-height ratio.

as a measure of stiffness. Some studies have described a greater association with measures of central or visceral adiposity in patients with diabetes and a general population.[3 13 17–19 21] However, Strasser et al[62] found no association with BF%; other studies found no association between BMI and cfPWV.[13 14] Rodrigues et al[15] reported that BMI was negatively associated with cfPWV (β=−0.103) in a large cohort.

The Whitehall II Cohort study[22] was completed based on staff lists from offices located in central London. It showed that all measures of adiposity were robust predictors of accelerated cfPWV after adjusting for potential confounding factors. The use of different measures to evaluate arterial stiffness such as CAVI and baPWV—as well as the emphasis on a population with intermediate cardiovascular risk—could explain some of the discrepancies with our study.

Arterial stiffness depends on arterial wall elasticity and diameter. A positive correlation was found between BMI and aortic diameter as measured by nuclear magnetic resonance.[63] This could partially explain the negative association between measures of adiposity and arterial stiffness. The different results obtained here can be explained in part by different methods of arterial stiffness measurements and the adjustment variables used. It might also be because CAVI is a measure of central and peripheral stiffness. In addition, the blood pressure at the time of the measurement does not seem to modify its value.[37 64 65] Conversely, the baPWV reflects peripheral arterial stiffness.[38] Other influences potentially underlying the observed differences are age, sex, race, prevalent cardiovascular diseases and drugs used for treatment of hypertension, diabetes mellitus and dyslipidaemia.[12 51 59 61] These differences between CAVI and baPWV are measures of rigidity and could explain these results suggesting a greater association of adiposity measurements with CAVI than with baPWV.

The proportion of baPWV variability explained by adiposity measurements in our study was less than 1% (between 0.7% for BMI and 0.1% for CUN-BAE). The results are also lower than those published for the general population by Wohlfahrt et al (5% for WHtR and 3% for BMI)[3] but are comparable with the results described by Rodrigues et al (BMI 0.7%).[15] The proportion of CAVI variability explained by adiposity measures was higher than 5% with CUN-BAE and BMI and higher than 3.5% with WHtR and BRI. In the subgroup analysis, the proportion of CAVI variability explained by adiposity measures was higher in diabetic, obese, younger, and non-hypertensive subjects. Our results show that the influence of adiposity measurements on CAVI is greater than on baPWV. To the best of our knowledge, no other study has yet analysed this aspect using CAVI. The novel results of this study may have important clinical relevance because they show the associations of both general and abdominal obesity measures with CAVI and baPWV in subjects with intermediate cardiovascular risk. Furthermore, the

**Table 3** Multiple regression analysis: association between adiposity measures with CAVI and baPWV

| | R² | Standardised β | No standardised β (95% CI) | Partial R² | p Value |
|---|---|---|---|---|---|
| | | | **CAVI** | | |
| BMI | 0.412 | −0.289 | −0.075 (−0.083 to −0.066) | 0.055 | <0.001 |
| WHtR | 0.381 | −0.221 | −3.650 (−4.207 to −3.093) | 0.038 | <0.001 |
| CUN-BAE | 0.410 | −0.450 | −0.069 (−0.077 to −0.061) | 0.058 | <0.001 |
| BRI | 0.378 | −0.215 | −0.142 (−0.165 to −0.120) | 0.037 | <0.001 |
| | | | **baPWV** | | |
| BMI | 0.402 | −0.100 | −0.057 (−0.076 to −0.038) | 0.007 | <0.001 |
| WHtR | 0.394 | −0.044 | −1.557 (−2.748 to −0.366) | 0.001 | 0.021 |
| CUN-BAE | 0.401 | −0.152 | −0.050 (−0.068 to −0.033) | 0.007 | <0.001 |
| BRI | 0.394 | −0.046 | −0.066 (−0.115 to −0.018) | 0.002 | 0.014 |

Four different multiple linear regression models were used to analyse the associations of adiposity measures with CAVI and baPWV.Adjusted for age (years), gender (0=male and 1=female), systolic blood pressure, smoking (0=no and 1=yes), atherogenic index, HbA1c and METs/min/week.baPWV, brachial-ankle pulse wave velocity; BMI, body mass index; BRI, body roundness index; CAVI, cardio-ankle vascular index; CUN-BAE, Clínica Universidad de Navarra-body adiposity estimator; METs-min/week, metabolic equivalent min per week; WHtR, waist-to-height ratio.

**Table 4** Association between adiposity measures with CAVI in different groups

| | R² | β (95% CI) | Partial R² | p Value |
|---|---|---|---|---|
| **CAVI** | | | | |
| Hypertensive | | | | |
| BMI | 0.409 | −0.076 (−0.087 to −0.065) | 0.023 | <0.001 |
| WHtR | 0.383 | −3.838 (−4.527 to −3.148) | 0.035 | <0.001 |
| CUN-BAE | 0.368 | −0.068 (−0.078 to −0.058) | 0.051 | <0.001 |
| BRI | 0.382 | −0.165 (−0.193 to −0.137) | 0.035 | <0.001 |
| Non-hypertensive | | | | |
| BMI | 0.372 | −0.071 (−0.086 to −0.057) | 0.052 | <0.001 |
| WHtR | 0.324 | −2.939 (−3.850 to −2.028) | 0.046 | <0.001 |
| CUN-BAE | 0.405 | −0.065 (−0.080 to −0.052) | 0.080 | <0.001 |
| BRI | 0.318 | −0.108 (−0.144 to −0.072) | 0.042 | <0.001 |
| Patients with diabetes | | | | |
| BMI | 0.439 | −0.087 (−0.101 to −0.073) | 0.95 | <0.001 |
| WHtR | 0.399 | −4.613 (−5.523 to −3.702) | 0.069 | <0.001 |
| CUN-BAE | 0.437 | −0.083 (−0.096 to −0.070) | 0.70 | <0.001 |
| BRI | 0.391 | −0.169 (−0.205 to −0.134) | 0.062 | <0.001 |
| Patients without diabetes | | | | |
| BMI | 0.396 | −0.067 (−0.078 to −0.056) | 0.049 | <0.001 |
| WHtR | 0.369 | −3.088 (−3.781 to −2.396) | 0.034 | <0.001 |
| CUN-BAE | 0.395 | −0.059 (−0.069 to −0.050) | 0.064 | <0.001 |
| BRI | 0.369 | −0.130 (−0.159 to −0.099) | 0.034 | 0.016 |
| Obese | | | | |
| BMI | 0.414 | −0.093 (−0.112 to −0.074) | 0.054 | <0.001 |
| WHtR | 0.377 | −2.945 (−4.085 to −1.804) | 0.003 | <0.001 |
| CUN-BAE | 0.418 | −0.100 (−0.122 to −0.079) | 0.026 | <0.001 |
| BRI | 0.377 | −0.110 (−0.153 to −0.069) | 0.003 | <0.001 |
| Non-obese | | | | |
| BMI | 0.373 | −0.058 (−0.078 to − 0.038) | 0.013 | <0.001 |
| WHtR | 0.361 | −1.394 (−2.366 to − 0.421) | 0.003 | 0.005 |
| CUN-BAE | 0.373 | −0.050 (−0.066 to −0.033) | 0.020 | <0.001 |
| BRI | 0.358 | −0.044 (−0.085 to −0.003) | 0.002 | 0.034 |
| ≤62 years | | | | |
| BMI | 0.368 | −0.078 (−0.089 to −0.077) | 0.77 | 0.003 |
| WHtR | 0.332 | −4.142 (−4.854 to −3.431) | 0.065 | <0.001 |
| CUN-BAE | 0.361 | −0.068 (−0.078 to −0.058) | 0.070 | <0.001 |
| BRI | 0.324 | −0.163 (−0.192 to −0.134) | 0.062 | <0.001 |
| >62 years | | | | |
| BMI | 0.234 | −0.074 (−0.088 to −0.060) | 0.056 | <0.001 |
| WHtR | 0.195 | −3.086 (−3.948 to −2.224) | 0.029 | <0.001 |
| CUN-BAE | 0.234 | −0.071 (−0.084 to −0.059) | 0.065 | <0.001 |
| BRI | 0.193 | −0.124 (−0.159 to −0.089) | 0.028 | <0.001 |
| Assets | | | | |
| BMI | 0.414 | −0.072 (−0.084 to −0.060) | 0.053 | <0.001 |
| WHtR | 0.392 | −3.570 (−4.335 to −2.805) | 0.034 | <0.001 |
| CUN-BAE | 0.413 | −0.066 (−0.075 to −0.053) | 0.060 | <0.001 |

**Table 4** Continued

|  | R² | β (95% CI) | Partial R² | p Value |
|---|---|---|---|---|
| BRI | 0.391 | −0.148 (−0.180 to −0.116) | 0.034 | <0.001 |
| Sedentary |  |  |  |  |
| BMI | 0.408 | −0.080 (−0.092 to −0.068) | 0.055 | <0.001 |
| WHtR | 0.364 | −3.776 (−4.598 to −2.954) | 0.045 | <0.001 |
| CUN-BAE | 0.404 | −0.074 (−0.086 to −0.062) | 0.059 | <0.001 |
| BRI | 0.360 | −0.144 (−0.176 to −0.111) | 0.043 | <0.001 |

Multiple linear regression models were used to analyse the associations of adiposity measures with CAVI by groups. Adjusted for age (years), gender (0=male and 1=female), systolic blood pressure, smoking (0=no and 1=yes), atherogenic index, HbA1c and METs/min/week. BMI, body mass index; BRI, body roundness index; CAVI, cardio-ankle vascular index; CUN-BAE, Clínica Universidad de Navarra-body adiposity estimator; METs-min/week, metabolic equivalent min per week; WHtR, waist-to-height ratio.

results provide information that could be used in new prospective studies and could potentially improve cardiovascular risk equations.

Various studies have analysed the effect of weight loss on arterial stiffness—most of these have been collected in two meta-analyses. The first analysed the results of 20 studies (1259 participants) and showed that losses of 8% of the weight after making changes in the diet and in the lifestyles do diminish the PWV. Diet and lifestyle interventions also seem to improve pulse wave velocity. The standardised mean difference (SMD) for the overall effect of weight loss on baPWV was −0.32 (p<0.001) cfPWV (SMD −0.35, p≤0.001) and baPWV (SMD −0.48, p<0.01); this improved with weight loss.[66] In the second meta-analysis, 43 studies (4231 participants) were included, and the average weight loss was 11% of the initial body weight; weight loss decreased CAVI (SMD=−0.48; p=0.04).[67]

In summary, our results show that the correlation between measures of adiposity and measures of arterial stiffness are greater with CAVI than with baPWV. The different measures of adiposity better explained the variability of arterial stiffness evaluated using CAVI than using baPWV. These suggest that the relationship with adiposity measures is greater if the arterial stiffness is measured using CAVI than with baPWV. This is likely because CAVI measures rigidity at the central and peripheral levels and is not affected by blood pressure at the time of measurement.[37 64 68 69]

The most important limitation of this study is its transversal design, which prevents the establishment of causal relationships as well as the direction of influence of adiposity measures on arterial stiffness. Another limitation is that the population was ethnically homogeneous (all subjects were Caucasians with intermediate cardiovascular risk). Therefore, the extrapolation of our findings may be limited.

## CONCLUSION

In conclusion, the adiposity measures analysed here show a negative association with arterial stiffness measures. The percentage of variation in CAVI that is explained by its relation to the different measures of adiposity ranges from 5.8% (CUN-BAE) to 3.7% (BRI). In the case of baPWV, it oscillates between 0.7% (CUN-BAE and BMI) and 0.1% (WHtR). These results suggest that measures of general adiposity and BF% better explain the variability of CAVI compared with measures of abdominal and visceral adiposity.

**Author affiliations**
[1]Primary Care Research Unit, The Alamedilla Health Center, Castilla and León Health Service (SACyL), Salamanca, Spain
[2]Biomedical Research Institute of Salamanca (IBSAL), Salamanca, Spain
[3]Department of Biomedical and Diagnostic Sciences, University of Salamanca, Salamanca, Spain
[4]Department of Statistics, University of Salamanca, Salamanca, Spain
[5]Department of Nursing and Physiotherapy, University of Salamanca, Salamanca, Spain
[6]San Agustín Health Center, Illes Balears Health Service (IBSALUT), Palma of Mallorca, Spain
[7]Institut Universitari d'Investigació en Atenció Primària Jordi Gol (IDIAP Jordi Gol), Girona, Spain
[8]Institut d'Investigació Biomèdica de Girona Dr. Josep Trueta (IDBGI), Girona, Spain
[9]Department of Medicine, University of Salamanca, Salamanca, Spain
[10]Departament de Ciències Mèdiques, Universitat de Girona, Girona, Spain

**Acknowledgements** We are grateful to all professionals participating in the MARK study. Lead author for this group: Rafel Ramos, Research Unit, Primary Health Care, Girona, Jordi Gol Institute for Primary Care Research (IDIAP Jordi Gol), Catalonia, Spain, email: rramos.girona.ics@gencat.net. Coordinating center: Rafel Ramos, Ruth Martí, Dídac Parramon, Anna Ponjoan, Miquel Quesada, Maria Garcia-Gil, Martina Sidera and Lourdes Camós, Research Unit, Primary Health Care, Jordi Gol Institute for Primary Care Research (IDIAP Jordi Gol), C/Maluquer Salvador, 11, 17002-Girona, Catalonia, Spain. Fernando Montesinos, Ignacio Montoya, Carlos López, Anna Agell, Núria Pagès of the Primary Care Services, Girona, Catalan Institute of Health (ICS), Catalonia, Spain. Irina Gil, Anna Maria-Castro of the Primary Care Services, Girona, Institut d'Assistència Sanitaria (IAS), Catalonia, Spain. Fernando Rigo, Guillermo Frontera, Antònia Rotger, Natalia Feuerbach, Susana Pons, Natividad Garcia, John Guillaumet, Micaela Llull and Mercedes Gutierrez of the Health Center Primary Care San Augustín, Ibsalut Balears, Spain. Cristina Agudo-Conde, Leticia Gómez-Sanchez, Carmen Castaño-Sanchez, Carmela Rodriguez-Martín, Benigna Sanchez-Salgado, Angela de Cabo-Laso, Emiliano Rodriguez-Sanchez, Jose Angel Maderuelo-Fernandez, Marta Gómez-Sánchez, Emilio Ramos-Delgado, Carmen Patino-Alonso, Jose I Recio-Rodriguez, Manuel A Gomez-Marcos and Luis Garcia-Ortiz, Primary Care Research Unit of The Alamedilla, Salamanca, Spain, Castilla and León Health Service-SACYL.

**Collaborators** MARK Group. redIAPP: Research Network in Preventive Activities and Health Promotion, Girona, Spain.

**Contributors** LG-S designed the study, wrote the protocol, participated in fundraising, interpreted the results, prepared the manuscript draft, performed all analytical testing, interpreted the results and reviewed the manuscript and corrected the final version of the manuscript. JIR-R, RM, FR and CA-C participated

in the study design, data collection and manuscript review. ER-S and JAM-F participated in the study design, interpretation of results and manuscript review. MCP-A participated in the analysis of results and final review of the manuscript. LG-O, RR and MAG-M participated in the protocol design, fundraising, analysis of results and final review of the manuscript. All authors reviewed and approved the final version of the manuscript.

**Funding** This research was supported by grants from the Spanish Ministry of Science and Innovation (MICINN), the Carlos III Health Institute/European Regional Development Fund (ERDF) (MICINN, ISCIII/FEDER) (Red RedIAPP RD06/0018), Research Groups: (RD16/0007/0003 (PI10/01088, PI10/02077, PI10/02043, PI13/01930) and the Regional Health Management of Castile and León (GRS 635/A/11; GRS 906/B/14).

**Competing interests** None declared.

**Patient consent** Obtained.

**Ethics approval** The study was approved by the Clinical Research Ethics Committee of the Primary Care Research Institute Jordi Gol, the Health Care Area of Salamanca and Palma of Mallorca.

**Provenance and peer review** Not commissioned; externally peer reviewed.

**Data sharing statement** The data used for the analysis of the results of this study are available and will be facilitated by the author.

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
