## [Reviewer comments · BMJ Open]

ARTICLE DETAILS

TITLE (PROVISIONAL)	Adiposity measures and arterial stiffness in primary care: the MARK prospective observational study
AUTHORS	Gomez-Sanchez, Leticia; Garcia-Ortiz, Luis; Patino-Alonso, Maria; Recio-Rodriguez, Jose; Frigo, Fernando; Marti, Rhut; Agudo-Conde, Cristina; Rodriguez-Sanchez, Emiliano; Maderuelo-Fernandez, Jose; Ramos, Rafel; Gomez-Marcos, Manuel

VERSION 1 - REVIEW

REVIEWER	Simon Rabkin University of British Columbia
REVIEW RETURNED	15-Mar-2017

GENERAL COMMENTS	The subject matter has been examined previously in a number of studies. There are ample data on adiposity and baPWV. There are little data on CAVI but the question is whether the use of this index of vascular stiffness contributes more precise or more novel information. The manuscript needs to define or discuss in greater detail why the authors believe that CAVI should be considered more reliable an indicator of vascular stiffness or relates more to adiposity. A major limitation of this study is the lack of data on exercise - frequency and intensity as exercise is important factor influencing arterial stiffness. It may have influenced the results if the subjects with greater obesity exercised less. There are longitudinal follow-up data (quoted by the authors) reported by Brunner et al in the Whitehall II cohort that adiposity in later midlife predicts accelerated progression of aortic stiffness. Longitudinal data is stronger than cross sectional data so the value of this study is less. This requires further discussion. Proof of causality is lacking in this kind of epidemiologic study. Studies on the impact of weight loss on arterial stiffness which is a good test of the question of causality has been conducted. Petersen et al summarized these data in a meta-analysis and concluded that "Modest weight loss (mean 8% of initial body weight) achieved with diet and lifestyle measures improved PWV" (Effect of weight loss on pulse wave velocity: systematic review and meta-analysis. Petersen KS; Blanch N; Keogh JB; Clifton PM. Arteriosclerosis, Thrombosis & Vascular Biology. 35(1):243-52, 2015). Considering that the clinical trials have already been done, what is the utility of an epidemiologic study of association? Specifics of the manuscript Figure 2 is confusing and could be deleted and discussed in the text.
---

REVIEWER	Masaaki Miyata Associate Professor, Kagoshima University, Japan
REVIEW RETURNED	30-Mar-2017

GENERAL COMMENTS	The authors demonstrated that adiposity measures are negatively associated with arterial stiffness measures in 2354 Caucasian adults with intermediate cardiovascular risk. This association was greater in CAVI compared with baPWV. The strength of this manuscript is that they use the several adiposity measures in Caucasian adults with intermediate cardiovascular risk. The weakness is the negative correlation between arterial stiffness and adiposity has been already reported in many previous manuscripts. Therefore, this reviewer thinks that the originality and priority of this manuscript are not enough high. Major comments  1. Please identify the machine which was used to measure the baPWV. As shown in Table 1, the gender difference was seen in CAVI, but not in baPWV. How do you comment about this difference? In addition, this reviewer would like to know the correlation between CAVI and baPWV in this study. 2. Total cholesterol levels > 250 mg/dl for the definition of dyslipidemia is not correct. The authors measured the LDL and HDL cholesterol levels and TG. Therefore, please use these values for the definition of dyslipidemia. 3. Page 9, line 2-19: Please delete these sentences, because they are repeated. 4. Page 11, line 1-3: This paragraph is difficult to understand. Minor comments  1. Page 4, line 17: The “populatio” should be “population”. 2. Page 11, line 3; (Table 1S) (supplementary material): Are these are correct? 3. Page 17: Some of Figure 2 legends are written in Spanish.
--

VERSION 1 – AUTHOR RESPONSE

Reviewer: 1

Simon Rabkin

University of British Columbia

Please state any competing interests or state 'None declared': None

The subject matter has been examined previously in a number of studies. There are ample data on adiposity and baPWV. There are little data on CAVI but the question is whether the use of this index of vascular stiffness contributes more precise or more novel information. The manuscript needs to define or discuss in greater detail why the authors believe that CAVI should be considered more reliable an indicator of vascular stiffness or relates more to adiposity.

Authors' Answer

As you said, there are many studies that have analyzed the relationship of the measures of adiposity with the pulse wave velocity (PWV) and few that have done so with the CAVI. The importance, relevance and novelty of this manuscript, as far as we know, is the relationship between these two measures of arterial stiffness, CAVI and baPWV, with several measures of adiposity, BMI, WHtR, CUN-BAE and the BRI. CAVI measures central and peripheral arterial stiffness and it is independent from the blood pressure at the time of measurement (1; 2; 3; 4). In contrast, baPWV is a parameter of

peripheral arterial stiffness, which is influenced by blood pressure in addition to the age. According to the results of this study, the correlation of adiposity measurements with arterial stiffness measurements is greater with CAVI than with baPWV. The different measures of adiposity would explain 4-5% of the CAVI variability, however, they would explain only 0.1-0.7% of the baPWV variability. Finally, as shown in Figure 1S, the mean CAVI values decrease as like as the values of the four adiposity measures increase, which does not happen with the baPWV. All this suggests that the association of the different measures of adiposity with the arterial stiffness is greater if we measure it with the CAVI than if we do it with the baPWV. In addition, the fact that the CAVI is a measure of arterial stiffness little used in western population highlights the originality of this study

We have introduced in the discussion of the manuscript the following sentence:

In summary, our results showed the correlation between measures of adiposity and measures of arterial stiffness is greater with CAVI than with baPWV. The different measures of adiposity better explain the variability of arterial stiffness evaluated using CAVI than using baPWV. All this suggests that the relationship with adiposity measures is greater if the arterial stiffness is measured using CAVI than using baPWV. This is likely because CAVI measures rigidity at the central and peripheral levels and it is not influenced by blood pressure at the time of measurement (1; 2; 3; 4). (Page 14, line 18).

A major limitation of this study is the lack of data on exercise - frequency and intensity as exercise is important factor influencing arterial stiffness. It may have influenced the results if the subjects with greater obesity exercised less.

Authors' Answer

Taking into account the limitations mentioned by the reviewer, and given that in the MARK study cohort we have collected the physical activity performed by the subjects, with the Minnesota LTPA Questionnaire (5) validated for Spanish men and women (6; 7). We have used physical activity, measured in metabolic equivalents (METs) minute week, as covariate adjustment in the regression analysis performed in table 3 and 4, and in the covariance analysis performed in Figure 1S. On the other hand, based on recommendations from the American Heart Association (8), we have considered as sedentary those participants who do not follow the recommendations of moderate-intensity aerobic PA practiced for a minimum of 30 min in 5 days per week (EEPA moderate < 675 kcal/week) or high-intensity aerobic PA practice for a minimum of 20 min in 3 days per week (EEPA intense < 420 kcal/week).

We have included the group of active and sedentary in the analysis performed by subgroups in table 4.

The results are practically unchanged, as can be seen in the new version of the manuscript.

We have made the following modifications in the manuscript:

ABSTRACT

Results

The proportion of CAVI variability explained by the adiposity measures were 5.5% for BMI, 5.8% for CUN-BAE, 3.8% for WHtR, and 3.7% for BRI, which were higher among diabetic, obese, younger (≤ 62 years), and non-hypertensive subjects who had similar activity and sedentary profiles. (Page 2, line 20).

METHODS

Variables and measurement instruments

Leisure time physical activity (LTPA) was collected using the Minnesota LTPA Questionnaire (5) that was validated for Spanish men and women(6; 7). The questionnaire was administered by trained interviewers who spent about 10 to 20 min per participant collecting detailed information about physical activity (PA) during the preceding year, the number of times this activity was performed, and the average duration of each activity on each occasion. Each PA has an intensity code, based on the ratio between the metabolic rate during PA practice and the basal metabolic rate (MET) (9). We assumed that 1 MET approximately corresponds to 1 kcal/min of energy expenditure. Therefore, we can calculate the total energy expenditure in leisure time of PA (EEPA total) in kilocalories per week.

Moreover, based on the PA intensity code, we could quantify the energy expenditure in physical activity (EEPA) according to the activity's classification as intense, moderate, or light intensity as follows: light PA intensity is below 4 METs, such as walking (EEPA light). Moderate PA intensity is 4–5.5 METs, such as brisk walking (EEPA moderate). Intense PA intensity is greater than or equal to 6 METs, such as jogging (EEPA intense). Thus, for each particular subject: EEPA total = EEPA light + EEPA moderate + EEPA intense. Based on recommendations from the American Heart Association (8), we considered those participants who do not meet the recommendations of moderate-intensity aerobic PA practice for a minimum of 30 min on 5 days each week (EEPA moderate < 675 kcal/week) or high-intensity aerobic PA practice for a minimum of 20 min on 3 days each week (EEPA intense < 420 kcal/week) to be sedentary. (Page 8, line 12).

Data analysis

All models were adjusted for..... METs/min/week(Page 10, line 2).

RESULTS

sedentariness (53.7 vs. 37.0). (Page 11, line 1).

In the multiple linearafter adjustment for.... METs/min/week. (Page 11, line 17).

Figure 1S (Supplementary material) shows the estimated marginal means of CAVI (a) and baPWV (b) by quartiles of the different adiposity measures. After adjustment for the variables used in the multiple linear regression analysis, the mean CAVI values decreased as the quartiles of the four adiposity measurements increased ($p < 0.05$). However, the same is not true of baPWV with WHtR and BRI ($p > 0.05$). (Page 11, line 10).

The proportions of CAVI variability explained in the respective multiple linear regression analyses by the adiposity measures were 5.5% for BMI, 5.8% for CUN-BAE, 3.8% for WHtR, and 3.7% for BRI. In the case of baPWV, the variance explained by the measures of adiposity were 0.7% for BMI, and CUN-BAE, and 0.1% for WHtR and 0.2 for BRI. (Page 11, line 17).

In the multiple linear regression analysis by subgroup, the proportion of CAVI variability explained by adiposity measures was higher among diabetics, obese, non-hypertensive, and subjects 62 years or younger, and was similar in active and sedentary people (Table 4). (Page 11, line 22).

Tables.

Table 1: We have included the variable physical activity (METs / min / week) as well as number and percentage of sedentary subjects.

Table 3: We have performed all the analysis again including in the models as adjustment variable the total physical activity performed by the subjects.

Table 4: We have performed all the analysis again including in the models as adjustment variable the total physical activity performed by the subjects, including the active and sedentary group.

DISCUSSION

The proportion of baPWV variability explained by adiposity measurements in our study was small than 1% (between 0.7% for BMI and 0.1% for CUN-BAE). (Page 14, line 5).

The proportion of CAVI variability explained by adiposity measures was higher than 5% with CUN-BAE and BMI and higher than 3.5% with WHtR and BRI. (Page 14, line 8).

There are longitudinal follow-up data (quoted by the authors) reported by Brunner et al in the Whitehall II cohort that adiposity in later midlife predicts accelerated progression of aortic stiffness. Longitudinal data is stronger than cross sectional data so the value of this study is less. This requires further discussion.

Authors' Answer

The Whitehall II Cohort study, is a prospective study including 10308 male and female civil servants (initially aged 35–55 years) recruited in 1985 to 1988. Adiposity measures were an independent predictor of accelerated arterial hardening. The Whitehall II Cohort study (10) found that all measures of general adiposity, central adiposity, and body fat percentage were predictors of accelerated arterial stiffness in adults.

There are some differences with our work that must be not forgotten: firstly, the measures of arterial stiffness used are different, in the Whitehall II Cohort study (10) it is used the cfPWV which measures stiffness in the aorta artery, a measure influenced by changes in blood pressure. However, in our study we have used the CAVI, that measures stiffness at the central and peripheral levels, which is independent from blood pressure at the time of measurement (1; 2; 3; 4), and on the other hand, the baPWV, which is a parameter of peripheral stiffness, influenced by blood pressure. Secondly, they analyze civil servants Londoners; while in our study we have analyzed Spanish subjects with intermediate cardiovascular risk. Thirdly, the percentage of classic cardiovascular risk factors and the percentage of subjects treated for control of blood pressure, diabetes and dyslipidemia was higher in our study. Finally, the mean age of the subjects in our study was 4 years less (61 years vs 65 years).

We have included in the discussion section the following sentence:

In the Whitehall II Cohort study (10), accomplished on the basis of staff lists from offices located in central London, it was found that all measures of adiposity were robust predictors of accelerated CFPWV, after adjusting them with potential confounding factors. The use of different measures to measure arterial stiffness such as CAVI and baPWV, as well as being a population with intermediate cardiovascular risk could explain some of the discrepancies with our study. (Page 13, line 13).

Proof of causality is lacking in this kind of epidemiologic study. Studies on the impact of weight loss on arterial stiffness which is a good test of the question of causality has been conducted. Petersen et al summarized these data in a meta-analysis and concluded that "Modest weight loss (mean 8% of initial body weight) achieved with diet and lifestyle measures improved PWV" (Effect of weight loss on pulse wave velocity: systematic review and meta-analysis. Petersen KS; Blanch N; Keogh JB; Clifton PM. *Arteriosclerosis, Thrombosis & Vascular Biology*. 35(1):243-52, 2015). Considering that the clinical trials have already been done, what is the utility of an epidemiologic study of association?

Authors' Answer

Cross-sectional studies allow us to generate hypotheses that we can later confirm in prospective studies. The different measures of adiposity reflect the state of health beyond the simple measurement of body weight. The usefulness of the association analysis between arterial stiffness and different measures of adiposity, and the capacity for explaining arterial stiffness for adiposity measurements is given because as far as we know, it has never been analyzed before in a Caucasian population at intermediate cardiovascular risk the association between the two measures of arterial stiffness, CAVI and baPWV, with four measures of adiposity, BMI, WHtR, CUN-BAE and BRI. Therefore, we think that what is really important in this manuscript is that the results obtained suggest that the CAVI, which is an easily performed stiffness measure, with high reproducibility, shows a better relationship than the baPWV with all the measures of stiffness used. On the other hand the results show that the CAVI would be better measure than the baPWV to evaluate the effect of the possible interventions on the distribution of the body adiposity or the simple decrease of weight, since it seems to provide a more consistent information.

Since it is a cross-sectional study, we can not establish causality and we have specified it in the following sections of the manuscript.

Strengths and limitations of this study

The main limitation of our study is its cross-sectional design, which does not allow establishment of causal relationships or the direction in which adiposity measures influence vascular function. (Page 2, line 13).

DISCUSSION

The main limitation of our study is its cross-sectional design, which does not allow us to establish causal relations or the direction of influence of adiposity measures in vascular function. (Page 15, line 1).

Specifics of the manuscript Figure 2 is confusing and could be deleted and discussed in the text.

Authors' Answer

We think that the figure provides a rapid visual information of the behavior of the two measures of stiffness as the quartiles of the adiposity measures increase. Because of this reason, we believe that it can be maintained as supplementary material. However, if you consider that it is better not to present it, you can leave only the commentary in the text of the manuscript.

In the present manuscript, after including in the covariance analysis the total physical activity as a support covariate, it would be as follows:

Figure 1S (Supplementary material) shows the estimated marginal means of CAVI (a) and baPWV (b) by quartiles of the different adiposity measures. After adjustment for the variables used in the multiple linear regression analysis, the mean CAVI values decreased as the quartiles of the four adiposity measurements increased ($p < 0.05$). However, the same is not true of baPWV with WHtR and BRI ($p > 0.05$). (Page 11, line 10).

Reviewer: 2 Masaaki Miyata

Associate Professor, Kagoshima University, Japan

Please state any competing interests or state 'None declared': None declared

Please leave your comments for the authors below

The authors demonstrated that adiposity measures are negatively associated with arterial stiffness measures in 2354 Caucasian adults with intermediate cardiovascular risk. This association was greater in CAVI compared with baPWV. The strength of this manuscript is that they use the several adiposity measures in Caucasian adults with intermediate cardiovascular risk. The weakness is the negative correlation between arterial stiffness and adiposity has been already reported in many previous manuscripts. Therefore, this reviewer thinks that the originality and priority of this manuscript are not enough high.

Authors' Answer

The importance, relevance and novelty of this manuscript, as far as we know, is the relationship between these two measures of arterial stiffness, CAVI and baPWV, with several measures of adiposity, BMI, WHtR, CUN-BAE and the BRI. CAVI measures central and peripheral arterial stiffness and it is independent from the blood pressure at the time of measurement (1; 2; 3; 4). In contrast, baPWV is a parameter of peripheral arterial stiffness, which is influenced by blood pressure in addition to the age.

According to the results of this study, the correlation of adiposity measurements with arterial stiffness measurements is greater with CAVI than with baPWV. The different measures of adiposity would explain 4-5% of the CAVI variability, however, they would explain only 0.1-0.7% of the baPWV variability. Finally, as shown in Figure 1S, the mean CAVI values decrease as like as the values of the four adiposity measures increase, which does not happen with the baPWV. All this suggests that the association of the different measures of adiposity with the arterial stiffness is greater if we measure it with the CAVI than if we do it with the baPWV. In addition, the fact that the CAVI is a measure of arterial stiffness little used in western population highlights the originality of this study

We have introduced in the discussion of the manuscript the following sentence:

In summary, our results showed the correlation between measures of adiposity and measures of arterial stiffness is greater with CAVI than with baPWV. The different measures of adiposity better explain the variability of arterial stiffness evaluated using CAVI than using baPWV. All this suggests that the relationship with adiposity measures is greater if the arterial stiffness is measured using CAVI than using baPWV. This is likely because CAVI measures rigidity at the central and peripheral levels and it is not influenced by blood pressure at the time of measurement (1; 2; 3; 4). (Page 14, line 18).

On the other hand we think that the association of arterial stiffness with adiposity measurements is not clear enough and we have reflected this in the introduction of the manuscript in with the following

phrases.

“However, the relation between adiposity and arterial stiffness remains controversial. Thus the body mass index (BMI) has been associated with arterial stiffness in the general population (11; 12), and in diabetic patients (13; 14). However, other works have not found this association (15), the association disappeared after adjusting for potential confounders (14), or showed a negative association (16; 17). On the other hand, there are studies that suggest a greater correlation of measures of central or visceral adiposity than measures of general adiposity with arterial stiffness in the general population (11; 18; 19; 20; 21), in diabetic patients (14), and in diabetics and hypertensive patients (22)”. (Page 4, line 10).

Major comments

1. Please identify the machine which was used to measure the baPWV. As shown in Table 1, the gender difference was seen in CAVI, but not in baPWV. How do you comment about this difference? In addition, this reviewer would like to know the correlation between CAVI and baPWV in this study.

Authors' Answer

We have added in the new version of the manuscript the data requested by the reviewer of the measurement device used to estimate baPWV and the correlation between CAVI and baPWV in this study.

The main differences observed in this manuscript between the two measures of stiffness used have been: The value of CAVI is higher in males than in females, and there are no differences in the case of the baPWV. There is evidence that CAVI linearly increases with age, and that this increase is greater in men than in women (approximately 0.2, equivalent to 4-5 years of age) (10; 23). Similarly, according to other cross-sectional data from 23257 healthy Japanese subjects (12729 males and 10528 females), the odds ratio (95% CI) for high CAVI (\geq 90th percentile) was 2.28 (2.06 - 2.54) in male (24). However, the influence of gender on PWV is not so clear, but if its relation with age and with blood pressure (25). Thus, a study that analyzed the influences of age and gender on the baPWV in 12517 subjects, found that the effect of age on baPWV differed according to gender (26). Women have a higher stiffness than men in prepubertal age, and increases after menopause. Men on the other hand experience a linear increase in arterial stiffness from puberty, suggesting that women have large arteries intrinsically more rigid than men, but these effects are mitigated by sex steroids during the reproductive years (27; 28).

We have made the following changes in the manuscript:

RESULTS

The mean value of CAVI was 8.8 ± 1.2 (8.9 in males and 8.6 in females, $p < 0.001$). (Page 10, line 12). The correlation between CAVI and baPWV was $r = 0.745$, ($p < 0.001$). (Page 11, line 3).

METHODS

CAVI was calculated using the VaSera VS-1500® device (Fukuda Denshi), and with the values obtained, the baPWV was estimated using the equation $baPWV = (0.5934 \times \text{height (cm)} + 14.4724) / tba$ (where tba is the time interval between the arm and ankle waves) (29). (Page 7, line 8).

DISCUSSION

In this study the mean value of CAVI was higher in males, which is in agreement with published data indicating that CAVI increases linearly with age, and is higher in males than in females (approximately 0.2, which is equivalent to 4–5 years old) (10; 23). Similarly, there was no difference in the mean baPWV between the sexes, which is consistent with data published by Tomiyama et al. (26) who showed that the effect of age on baPWV is different according to sex. Females have a higher arterial stiffness than prepubertal males, and this increases after menopause. Men, however, experience a linear increase in arterial stiffness from puberty. This suggests that women have large arteries that are intrinsically more rigid compared with men, but these effects are mitigated by sex steroids during the reproductive years (27; 28). (Page 12, line 7).

2. Total cholesterol levels > 250 mg/dl for the definition of dyslipidemia is not correct. The authors measured the LDL and HDL cholesterol levels and TG. Therefore, please use these values for the definition of dyslipidemia.

Authors' Answer

Dyslipidemia was defined if they were treated with lipid-lowering drugs or had altered LDL \geq 130 mg/dl, HDL \leq 45 mg/dl in men and \leq 55 in women, and TG \geq 150 mg/dl, as established by the European Society of Cardiology (ESC) and the European Atherosclerosis Society (EAS) 2011 (30). We have made the following changes in the version of the manuscript:

METHODS

Dyslipidemia was defined if they were treated with lipid-lowering drugs or had altered LDL \geq 130 mg/dl, HDL \leq 45 mg/dl in men and \leq 55 in women, and TG \geq 150 mg/dl, as established by the European Society of Cardiology (ESC) and the European Atherosclerosis Society (EAS) 2011 (30). (Page 7, line 20).

Table 1: we modified the number and percentage of subjects with dyslipidemia.

3. Page 9, line 2-19: Please delete these sentences, because they are repeated.

Authors' Answer

In the new version of the manuscript we have eliminated repeated sentences.

4. Page 11, line 1-3: This paragraph is difficult to understand.

Authors' Answer

In the new version we have modified the wording:

In the multiple linear regression analysis by subgroup, the proportion of CAVI variability explained by adiposity measures was higher among diabetics, obese, non-hypertensive, subjects 62 years or younger and was similar in actives and sedentary (Table 4). (Page 11, line 22).

Minor comments

1. Page 4, line 17: The "populatio" should be "population".

Authors' Answer

We have corrected the spelling mistake.

2. Page 11, line 3; (Table 1S) (supplementary material): Are these are correct?

Authors' Answer

We have fixed the error remaining in the current version:

In the multiple linear regression analysis by subgroup, the proportion of CAVI variability explained by adiposity measures was higher among diabetics, obese, non-hypertensive subjects, and age 62 years or younger and was similar in actives and sedentary (Table 4). (Page 11, line 22).

3. Page 17: Some of Figure 2 legends are written in Spanish.

Authors' Answer

We have edited the manuscript correcting all misspellings. Following the recommendations of the first reviewer figure 2 we have gone to supplementary material and left in the current version the following sentence:

Figure 1S (Supplementary material) shows the estimated marginal means of CAVI (a) and baPWV (b) by quartiles of the different adiposity measures. After adjustment for the variables used in the multiple linear regression analysis, the mean CAVI values decreased as the quartiles of the four adiposity measurements increased ($p < 0.05$). However, the same is not true of baPWV with WHtR and BRI ($p > 0.05$). (Page 11, line 10).

REFERENCES

1. Kubozono T, Miyata M, Ueyama K et al. (2007) Clinical significance and reproducibility of new

- arterial distensibility index. *Circ J* 71, 89-94.
2. Nagayama D, Endo K, Ohira M et al. (2013) Effects of body weight reduction on cardio-ankle vascular index (CAVI). *Obes Res Clin Pract* 7, e139-e145.
 3. Shirai K, Hiruta N, Song M et al. (2011) Cardio-ankle vascular index (CAVI) as a novel indicator of arterial stiffness: theory, evidence and perspectives. *J Atheroscler Thromb* 18, 924-938.
 4. Takaki A, Ogawa H, Wakeyama T et al. (2008) Cardio-ankle vascular index is superior to brachial-ankle pulse wave velocity as an index of arterial stiffness. *Hypertens Res* 31, 1347-1355.
 5. Taylor HL, Jacobs DR, Jr., Schucker B et al. (1978) A questionnaire for the assessment of leisure time physical activities. *J Chronic Dis* 31, 741-755.
 6. Elosua R, Marrugat J, Molina L et al. (1994) Validation of the Minnesota Leisure Time Physical Activity Questionnaire in Spanish men. The MARATHOM Investigators. *Am J Epidemiol* 139, 1197-1209.
 7. Elosua R, Garcia M, Aguilar A et al. (2000) Validation of the Minnesota Leisure Time Physical Activity Questionnaire In Spanish Women. Investigators of the MARATDON Group. *Med Sci Sports Exerc* 32, 1431-1437.
 8. Haskell WL, Lee IM, Pate RR et al. (2007) Physical activity and public health: updated recommendation for adults from the American College of Sports Medicine and the American Heart Association. *Circulation* 116, 1081-1093.
 9. Ainsworth BE, Haskell WL, Leon AS et al. (1993) Compendium of physical activities: classification of energy costs of human physical activities. *Med Sci Sports Exerc* 25, 71-80.
 10. Brunner EJ, Shipley MJ, Ahmadi-Abhari S et al. (2015) Adiposity, obesity, and arterial aging: longitudinal study of aortic stiffness in the Whitehall II cohort. *Hypertension* 66, 294-300.
 11. Wohlfahrt P, Somers VK, Cifkova R et al. (2014) Relationship between measures of central and general adiposity with aortic stiffness in the general population. *Atherosclerosis* 235, 625-631.
 12. Wohlfahrt P, Krajcoviechova A, Seidlerova J et al. (2013) Lower-extremity arterial stiffness vs. aortic stiffness in the general population. *Hypertens Res* 36, 718-724.
 13. Moh MC, Sum CF, Lam BC et al. (2015) Evaluation of body adiposity index as a predictor of aortic stiffness in multi-ethnic Asian population with type 2 diabetes. *Diab Vasc Dis Res* 12, 111-118.
 14. Teoh WL, Price JF, Williamson RM et al. (2013) Metabolic parameters associated with arterial stiffness in older adults with Type 2 diabetes: the Edinburgh Type 2 diabetes study. *J Hypertens* 31, 1010-1017.
 15. Hansen TW, Jeppesen J, Rasmussen S et al. (2004) Relation between insulin and aortic stiffness: a population-based study. *J Hum Hypertens* 18, 1-7.
 16. Rodrigues SL, Baldo MP, Lani L et al. (2012) Body mass index is not independently associated with increased aortic stiffness in a Brazilian population. *Am J Hypertens* 25, 1064-1069.
 17. Huisman HW, Schutte R, Venter HL et al. (2015) Low BMI is inversely associated with arterial stiffness in Africans. *Br J Nutr* 113, 1621-1627.
 18. Canepa M, AlGhatrif M, Pestelli G et al. (2014) Impact of central obesity on the estimation of carotid-femoral pulse wave velocity. *Am J Hypertens* 27, 1209-1217.
 19. Scuteri A, Orru M, Morrell CH et al. (2012) Associations of large artery structure and function with adiposity: effects of age, gender, and hypertension. The SardiNIA Study. *Atherosclerosis* 221, 189-197.
 20. Johansen NB, Vistisen D, Brunner EJ et al. (2012) Determinants of aortic stiffness: 16-year follow-up of the Whitehall II study. *PLoS One* 7, e37165.
 21. Wildman RP, Mackey RH, Bostom A et al. (2003) Measures of obesity are associated with vascular stiffness in young and older adults. *Hypertension* 42, 468-473.
 22. Recio-Rodriguez JI, Gomez-Marcos MA, Patino-Alonso MC et al. (2012) Abdominal obesity vs general obesity for identifying arterial stiffness, subclinical atherosclerosis and wave reflection in healthy, diabetics and hypertensive. *BMC Cardiovasc Disord* 12, 3.
 23. Choi SY, Oh BH, Bae Park J et al. (2013) Age-associated increase in arterial stiffness measured according to the cardio-ankle vascular index without blood pressure changes in healthy adults. *J Atheroscler Thromb* 20, 911-923.

24. Nagayama D, Watanabe R, Watanabe Y et al. (2016) OS 10-04 INVERSE RELATIONSHIP BETWEEN CARDIO-ANKLE VASCULAR INDEX (CAVI) AND BODY MASS INDEX IN HEALTHY JAPANESE SUBJECTS: A CROSS-SECTIONAL STUDY. J Hypertens 34 Suppl 1 - ISH 2016 Abstract Book, e73.
25. Coutinho T (2014) Arterial stiffness and its clinical implications in women. Can J Cardiol 30, 756-764.
26. Tomiyama H, Yamashina A, Arai T et al. (2003) Influences of age and gender on results of noninvasive brachial-ankle pulse wave velocity measurement--a survey of 12517 subjects. Atherosclerosis 166, 303-309.
27. Marlatt KL, Kelly AS, Steinberger J et al. (2013) The influence of gender on carotid artery compliance and distensibility in children and adults. J Clin Ultrasound 41, 340-346.
28. Bartok CJ, Marini ME, Birch LL (2011) High body mass index percentile accurately reflects excess adiposity in white girls. J Am Diet Assoc 111, 437-441.
29. Yamashina A, Tomiyama H, Takeda K et al. (2002) Validity, reproducibility, and clinical significance of noninvasive brachial-ankle pulse wave velocity measurement. Hypertens Res 25, 359-364.
30. Catapano AL, Reiner Z, De Backer G et al. (2011) ESC/EAS Guidelines for the management of dyslipidaemias The Task Force for the management of dyslipidaemias of the European Society of Cardiology (ESC) and the European Atherosclerosis Society (EAS). Atherosclerosis 217, 3-46.

VERSION 2 – REVIEW

REVIEWER	Professor Simon W Rabkin University of British Columbia Vancouver Canada
REVIEW RETURNED	13-May-2017

GENERAL COMMENTS	This observational study examines the association between obesity and arterial stiffness. The authors contend as is written several times – abstract and conclusion that “The arterial stiffness variability is better explained for BMI and CUN-BAE and when CAVI is used as a measure of stiffness rather than baPWV.” (from the abstract) This statement and phraseology and concept needs to be corrected or additional statistical analysis needs to be performed in order to support this contention.. (i) There is no testing that the correlation of an obesity index with one measure of arterial stiffness is significantly greater with one correlation than the other (ii) Table 3 shows that for the index of obesity WhtR, the correlation coefficient is higher for baPWV than for CAVI. For the index of obesity BRI, the r value is higher for baPWV than it is for CAVI. Thus Table 3 which is the multivariate analysis refutes the author’s claims. They cannot discard these findings and make the major claim that suggests that CAVI is better as they use the two indexes BMI and CUN-BAE and ignore WhtR and BRI. Alternatively they need to prove that WhtR and BRI are not good indices of obesity. The authors use the wording “arterial stiffness variability is better explained ... The word 'explained" suggests that there is causal relationship between obesity and arterial stiffness. While this may be the case, their study only examined an association. In reality there is a minimally higher correlation (association) for some indexes of obesity and not for others. This terminology must be changed.
---

	The authors do not include a scientific explanation for why CAVI should be better than BaPWV. The entry criteria into the study needs to be outlines better There is no discussion of clinical trials on the effect of weight loss on changes to arterial stiffness. These kind of data are of critical important to establish causality. These studies must be included in the discussion section. Minor points The abstract should make it more obvious how they got the abbreviation CUN-BAE The introduction needs to be rewritten to make it flow better rather than appear to have disjointed sentences The methods section should consistently use the past tense Final paragraph: correct the typo baPWW should be baPWV
--	---

VERSION 2 – AUTHOR RESPONSE

Reviewer(s)' Comments to Author:

Reviewer: 1 Reviewer: Professor Simon W Rabkin University of British Columbia, Vancouver, Canada

Please state any competing interests or state 'None declared': No competing interests

This observational study examines the association between obesity and arterial stiffness.

The authors contend as is written several times – abstract and conclusion that “The arterial stiffness variability is better explained for BMI and CUN-BAE and when CAVI is used as a measure of stiffness rather than baPWV.” (from the abstract) This statement and phraseology and concept needs to be corrected or additional statistical analysis needs to be performed in order to support this contention.

Authors' Answer

We have modified the summary and conclusion as follows:

Adiposity measures are negatively associated with arterial stiffness measures. The percentage of variation in CAVI explained by its relation to the different measures of adiposity ranges from 5.8% (CUN-BAE) to 3.7% (BRI). In the case of baPWV it oscillates between 0.7% (CUN-BAE and BMI) and 0.1% (WHtR). (Page 2, line 24; Page 16, line 6).

(I) There is no testing that the correlation of an obesity index with one measure of arterial stiffness is significantly greater with one correlation than the other.

Authors' Answer

As can be seen in Table 1S, there are significant differences between the CAVI and baPWV correlation coefficients with adiposity measures assessed, adjusted and unadjusted, using Stiger's Z test. The table is shown at the end of these answers, and if it is considered to be necessary it can be included as Table 1S (Supplementary material).

We have included the following sentence in the statistical analysis section.

We used Steiger's Z statistics for testing the significance of the difference between correlations

coefficients (1). (Page 10, line 1).

We have included the following sentence in the results section and in the legend of table 2.

Results

We found differences in correlation coefficients between CAVI, baPWV and measures of adiposity ($p < 0.001$ in all cases). (Page 11, line 15).

Legend of table 2

The correlation coefficients between CAVI, baPWV and adiposity measurements showed significant differences ($p < 0.001$ in all cases). (Page 25, line 2).

(II) Table 3 shows that for the index of obesity WhtR, the correlation coefficient is higher for baPWV than for CAVI. For the index of obesity BRI, the r value is higher for baPWV than it is for CAVI. Thus Table 3 which is the multivariate analysis refutes the author's claims. They cannot discard these findings and make the major claim that suggests that CAVI is better as they use the two indexes BMI and CUN-BAE and ignore WhtR and BRI. Alternatively they need to prove that WhtR and BRI are not good indices of obesity.

Authors' Answer

We think that the first question posed by the reviewer refers to table 2, table that shows the correlation coefficient between CAVI and baPWV with measures of adiposity.

Table 2 shows that the correlation coefficient adjusted for the index of obesity WhtR with CAVI is $r = -0.222$ ($p < 0.01$) and with baPWV is $r = 0.001$ ($p > 0.05$). The correlation coefficient adjusted for the index of obesity BRI with CAVI is $r = -0.218$ ($p < 0.01$) and with baPWV is $r = 0.005$ ($p > 0.05$). These results indicate that CAVI has a negative relation with WhtR and BRI. However, in the case of baPWV the correlation coefficient does not reach statistical significance, neither with WhtR nor with BRI.

We have included the following sentence in the results section and in the legend of table 2.

Results

We found differences in correlation coefficients between CAVI, baPWV and measures of adiposity ($p < 0.001$ in all cases). (Page 11, line 15).

Legend of table 2

The correlation coefficients between CAVI baPWV, and adiposity measurements showed significant differences ($p < 0.001$ in all cases). (Page 25, line 2).

For a better interpretation of the relationship between adiposity measurements, CAVI and baPWV we have calculated the standardized β coefficients based on the typical scores, and therefore directly compared between each other. The results of the standardized β coefficient have been added to Table 3.

According to these results, the negative association is greater when the different measures of adiposity are compared with CAVI than with baPWV, in all cases.

We have included the following sentence in the results section and we added in table 3 the standardized β coefficients.

The association between adiposity measurements and CAVI revealed standardized β between -0.450 (CUN-BAE) and -0.221 (WhtR). In the case of baPWV the values oscillate between -0.152 (CUN-BAE) and -0.044 (WhtR). (Page 12, line 4 and table 3).

The authors use the wording “arterial stiffness variability is better explained. The word 'explained" suggests that there is causal relationship between obesity and arterial stiffness. While this may be the case, their study only examined an association. In reality there is a minimally higher correlation (association) for some indexes of obesity and not for others. This terminology must be changed.

Authors' Answer

We have made the following changes.

This study investigates the relationship between adiposity measures and arterial stiffness in Caucasian adults with intermediate cardiovascular risk. (Page 2, line 4).

The secondary aim was to analyze the differences between the association of adiposity measures with CAVI and with baPWV. (Page 5, line 2).

The proportion of CAVI variability which can be attributed to the variation in the adiposity measures was 5.5% for BMI, 5.8% for CUN-BAE, 3.8% for WHtR, and 3.7% for BRI. (Page 11, line 25).

The results of this study show that adiposity measures have a negative association with arterial stiffness, especially CAVI. BMI and CUN-BAE are the ones with the highest coefficient of determination. (Page 12, line 14).

The authors do not include a scientific explanation for why CAVI should be better than BaPWV.

Authors' Answer

We have completed the discussion section where we talk about the differences between CAVI and baPWV:

These discrepancies between studies may be partially explained by different methods of arterial stiffness measurement and the adjustment variables used. This may be because CAVI reflects central and peripheral arterial stiffness and is less influenced by blood pressure values at the time of measurement (2; 3; 4). Conversely, arterial stiffness assessed using baPWV is a measure of peripheral arterial stiffness (5). Other potential influences on the observed differences are age, sex, race, prevalent cardiovascular risk, and drugs used for treatment of the different risk factor (6; 7; 8; 9). These differences between CAVI and baPWV, as measures of rigidity, could explain the results shown in this study, suggesting a greater association of adiposity measurements with CAVI than with baPWV. (Page 14, line 13).

The entry criteria into the study needs to be outlined better

Authors' Answer

Following the indications of the reviewer we have expanded the information in the current manuscript, remaining in the current version as follows:

This study analyzed 2354 of the 2495 subjects recruited in the MARK study. For the present analysis, we excluded 141 individuals, with $ABI \leq 0.9$ (n=99), or which CAVI (n=16), baPWV (n=12) and WC (n = 14) measurements were incomplete (Figure 1). (Page 5, line 26).

We have also included the information collected in the Flow Chart as reflected in Figure 1.

There is no discussion of clinical trials on the effect of weight loss on changes to arterial stiffness. These kind of data are of critical important to establish causality. These studies must be included in

the discussion section.

Authors' Answer

We have introduced the following paragraph in the discussion.

Numerous studies have been carried out analyzing the effect of weight loss on arterial stiffness, most of which have been collected in two meta-analyses. The first analyzed the results of 20 studies (1259 participants), and showed that modest weight loss (8% of initial body weight) achieved with diet and lifestyle interventions seems to improve PWV. The standardized mean difference (SMD) for the overall effect of weight loss on baPWV measured at all sites was -0.32 (95% CI, -0.41, -0.24; $P=0.0001$). cfPWV (SMD, -0.35; 95% CI, -0.44, -0.26; $P=0.0001$; 16 studies) and baPWV (SMD, -0.48; 95% CI, -0.78, -0.18; $P=0.002$; 5 studies) improved with weight loss. (10). In the second meta-analysis, 43 studies (4231 participants) were included and it was found that the average weight loss was approximately 11% of the initial body weight and that weight loss improved CAVI (SMD= -0.48; $p = 0.04$) (11). (Page 15, line 5)

Minor comments

The abstract should make it more obvious how they got the abbreviation CUN-BAE

Authors' Answer

We have modified the meaning of the abbreviation in the abstract, anthropometric measurements and foot of the tables, CUN-BAE as published in the original manuscript (Clínica Universidad de Navarra-Body Adiposity Estimator) (12). (Page 2, line 10; Page 6, line 18 and in tables).

The introduction needs to be rewritten to make it flow better rather than appear to have disjointed sentences

Authors' Answer

We have revised the introduction of the manuscript and joined different paragraphs so that the reading of the manuscript is more fluid.

The introduction in the new version is as follows.

INTRODUCTION

Obesity has been linked to increased all-cause and cardiovascular mortality (13). However, the mechanisms through which obesity can increase the frequency of cardiovascular disease beyond traditional risk factors are not clearly identified (14). It has been suggested that, increased arterial stiffness may be a mechanism by which obesity increases cardiovascular risk independently of traditional risk factors(15). It is known that arterial stiffness as evaluated by the brachial ankle pulse wave velocity (baPWV) is an independent predictor of coronary heart disease and mortality in both the general population (16) and in patients with diabetes mellitus (17; 18). Similarly, the cardio-ankle vascular index (CAVI) is associated with carotid and coronary atherosclerosis (19; 20; 21) and is a predictor of cardiovascular events in obese patients (22). I. However, the relationship between adiposity and arterial stiffness remains controversial. In this respect, there are studies that show that the body mass index (BMI) has been associated with arterial stiffness in the general population (15; 23) and in diabetic patients (6; 24). However, other research has not found this association (25), or the association disappeared after adjusting for potential confounders (24), or it showed a negative association (26; 27). Additionally, there are studies that suggest a stronger correlation of measures of

central or visceral adiposity than measures of general adiposity with arterial stiffness in the general population (15; 28; 29; 30; 31), in diabetic patients (24), and in diabetics and hypertensive patients (32). Finally the Whitehall II Cohort study (33) showed that all measures of general adiposity, central adiposity, and body fat percentage were predictors of accelerated arterial stiffness in adults. In this context the analysis of the relationship between arterial stiffness and different measures of adiposity can help to understand the role of obesity in cardiovascular disease.

Our study was designed bearing in mind that cardiovascular events are more likely to occur in patients with intermediate cardiovascular risk (34), and that there is a lack of studies analyzing the relationship of different adiposity measures with arterial stiffness in these subjects. We have established the following objectives: the primary aim of this study was to investigate the relationship between adiposity measures and arterial stiffness in Caucasian adults with intermediate cardiovascular risk. The secondary aim was to analyze the differences between the associations of adiposity measures with distinct arterial stiffness markers. (Page 4 and 5).

The methods section should consistently use the past tense

Authors' Answer

We have reviewed the methods section and put all the verbal tenses in the past.

This trial was a cross-sectional study of subjects recruited to the improving interMediate Risk management (MARK) study (NCT01428934) (35), which was a longitudinal study designed to assess whether the ankle-brachial index, arterial stiffness (measured by CAVI), postprandial glucose, glycosylated hemoglobin, self-measured blood pressure, and the presence of comorbidities were independently associated with the occurrence of vascular events. (Page 5, line 7).

It also investigated whether the predictive capacity of current risk equations could be improved in the intermediate risk population. The current study focused on the baseline visit. (Page 5, line 12).

CAVI was measured using a VaSera VS-1500® device (Fukuda Denshi) (36; 37). CAVI values were calculated automatically by estimating the stiffness parameter β using the following equation: $\beta = 2\rho \times 1 / (P_s - P_d) \times \ln(P_s / P_d) \times PWV^2$, where ρ was blood density, P_s and P_d were SBP and DBP in mmHg, and PWV was measured between the aortic valve and the ankle (3). The mean coefficient of variation of CAVI measurement was less than 5%, which was small enough to allow for clinical use of the index and confirmed that CAVI was reproducible index (37). (Page 7, line 7).

CAVI was calculated using the VaSera VS-1500® device (Fukuda Denshi), and with the values obtained, the baPWV was estimated using the equation $baPWV = (0.5934 \times \text{height (cm)} + 14.4724) / tba$ (where tba was the time interval between the arm and ankle waves) (5). (Page 7, line 10).

Each PA had an intensity code, based on the ratio between the metabolic rate during PA practice and the basal metabolic rate (MET). (Page 8, line 22).

It is assumed that 1 MET corresponds to approximately 1 kcal/min of energy expenditure. Therefore, we could calculate the total energy expenditure in leisure time of PA (EEPA total) in kilocalories per week. Moreover, based on the PA intensity code, we could quantify the energy expenditure in physical activity (EEPA) according to the activity's classification as intense, moderate, or light intensity as follows: light PA intensity was below 4 METs, such as walking (EEPA light). Moderate PA intensity was 4–5.5 METs, such as brisk walking (EEPA moderate). Intense PA intensity was greater than or equal to 6 METs, such as jogging (EEPA intense). Thus, for each particular subject: $EEPA\ total = EEPA\ light + EEPA\ moderate + EEPA\ intense$. (Page 8, line 24).

Final paragraph: correct the typo baPWW should be baPWV

Authors' Answer

We have corrected the error by remaining in the new version of the baPWV manuscript. (Page 15, line 22).

Table 1S (Supplementary material): Differences in correlation coefficients using Steiger's Z test between CAVI and baPWV.

Unadjusted Adjusted *

Z p Z p

BMI 15.9 <0.001 16.5 <0.001

WHtR 14.3 <0.001 15.4 <0.001

CUN-BAE 16.6 <0.001 25.2 <0.001

BRI 13.9 <0.001 15.4 <0.001

Abbreviations: CAVI, cardio-ankle vascular index. baPWV, brachial-ankle pulse wave velocity. BMI, body mass index. WHtR, waist-to-height ratio. CUN-BAE, Clínica Universidad de Navarra-body adiposity estimator. BRI, body roundness index.

*Adjusted for age, sex and systolic blood pressure.

p-values of differences in correlation coefficients

REFERENCES

1. Steiger JH (1980) Tests for comparing elements of a correlation matrix. *Psychological Bulletin* 87,, 245-251.
2. Takaki A, Ogawa H, Wakeyama T et al. (2008) Cardio-ankle vascular index is superior to brachial-ankle pulse wave velocity as an index of arterial stiffness. *Hypertens Res* 31, 1347-1355.
3. Shirai K, Hiruta N, Song M et al. (2011) Cardio-ankle vascular index (CAVI) as a novel indicator of arterial stiffness: theory, evidence and perspectives. *J Atheroscler Thromb* 18, 924-938.
4. Shirai K (2011) Analysis of vascular function using the cardio-ankle vascular index (CAVI). *Hypertens Res* 34, 684-685.
5. Yamashina A, Tomiyama H, Takeda K et al. (2002) Validity, reproducibility, and clinical significance of noninvasive brachial-ankle pulse wave velocity measurement. *Hypertens Res* 25, 359-364.
6. Moh MC, Sum CF, Lam BC et al. (2015) Evaluation of body adiposity index as a predictor of aortic stiffness in multi-ethnic Asian population with type 2 diabetes. *Diab Vasc Dis Res* 12, 111-118.
7. Kim HL, Lee JM, Seo JB et al. (2015) The effects of metabolic syndrome and its components on arterial stiffness in relation to gender. *J Cardiol* 65, 243-249.
8. Weng C, Yuan H, Yang K et al. (2013) Gender-specific association between the metabolic syndrome and arterial stiffness in 8,300 subjects. *Am J Med Sci* 346, 289-294.
9. Tomiyama H, Yamashina A, Arai T et al. (2003) Influences of age and gender on results of noninvasive brachial-ankle pulse wave velocity measurement--a survey of 12517 subjects. *Atherosclerosis* 166, 303-309.
10. Petersen KS, Blanch N, Keogh JB et al. (2015) Effect of weight loss on pulse wave velocity: systematic review and meta-analysis. *Arteriosclerosis, thrombosis, and vascular biology* 35, 243-252.
11. Petersen KS, Clifton PM, Lister N et al. (2016) Effect of weight loss induced by energy restriction on measures of arterial compliance: A systematic review and meta-analysis. *Atherosclerosis* 247, 7-20.
12. Gomez-Ambrosi J, Silva C, Catalan V et al. (2012) Clinical usefulness of a new equation for estimating body fat. *Diabetes Care* 35, 383-388.
13. Adams KF, Schatzkin A, Harris TB et al. (2006) Overweight, obesity, and mortality in a large prospective cohort of persons 50 to 71 years old. *N Engl J Med* 355, 763-778.
14. Poirier P, Giles TD, Bray GA et al. (2006) Obesity and cardiovascular disease: pathophysiology, evaluation, and effect of weight loss: an update of the 1997 American Heart Association Scientific

Statement on Obesity and Heart Disease from the Obesity Committee of the Council on Nutrition, Physical Activity, and Metabolism. *Circulation* 113, 898-918.

15. Wohlfahrt P, Somers VK, Cifkova R et al. (2014) Relationship between measures of central and general adiposity with aortic stiffness in the general population. *Atherosclerosis* 235, 625-631.
16. Turin TC, Kita Y, Rumana N et al. (2010) Brachial-ankle pulse wave velocity predicts all-cause mortality in the general population: findings from the Takashima study, Japan. *Hypertens Res* 33, 922-925.
17. Maeda Y, Inoguchi T, Etoh E et al. (2014) Brachial-ankle pulse wave velocity predicts all-cause mortality and cardiovascular events in patients with diabetes: the Kyushu Prevention Study of Atherosclerosis. *Diabetes Care* 37, 2383-2390.
18. Ikura K, Hanai K, Oka S et al. (2016) Brachial-ankle pulse wave velocity, but not ankle-brachial index, predicts all-cause mortality in patients with diabetes after lower extremity amputation. *J Diabetes Investig*.
19. Izuhara M, Shioji K, Kadota S et al. (2008) Relationship of cardio-ankle vascular index (CAVI) to carotid and coronary arteriosclerosis. *Circ J* 72, 1762-1767.
20. Okura T, Watanabe S, Kurata M et al. (2007) Relationship between cardio-ankle vascular index (CAVI) and carotid atherosclerosis in patients with essential hypertension. *Hypertens Res* 30, 335-340.
21. Nakamura K, Tomaru T, Yamamura S et al. (2008) Cardio-ankle vascular index is a candidate predictor of coronary atherosclerosis. *Circ J* 72, 598-604.
22. Satoh-Asahara N, Kotani K, Yamakage H et al. (2015) Cardio-ankle vascular index predicts for the incidence of cardiovascular events in obese patients: a multicenter prospective cohort study (Japan Obesity and Metabolic Syndrome Study: JOMS). *Atherosclerosis* 242, 461-468.
23. Wohlfahrt P, Krajcoviechova A, Seidlerova J et al. (2013) Lower-extremity arterial stiffness vs. aortic stiffness in the general population. *Hypertens Res* 36, 718-724.
24. Teoh WL, Price JF, Williamson RM et al. (2013) Metabolic parameters associated with arterial stiffness in older adults with Type 2 diabetes: the Edinburgh Type 2 diabetes study. *J Hypertens* 31, 1010-1017.
25. Hansen TW, Jeppesen J, Rasmussen S et al. (2004) Relation between insulin and aortic stiffness: a population-based study. *J Hum Hypertens* 18, 1-7.
26. Rodrigues SL, Baldo MP, Lani L et al. (2012) Body mass index is not independently associated with increased aortic stiffness in a Brazilian population. *Am J Hypertens* 25, 1064-1069.
27. Huisman HW, Schutte R, Venter HL et al. (2015) Low BMI is inversely associated with arterial stiffness in Africans. *Br J Nutr* 113, 1621-1627.
28. Canepa M, AlGhatrif M, Pestelli G et al. (2014) Impact of central obesity on the estimation of carotid-femoral pulse wave velocity. *Am J Hypertens* 27, 1209-1217.
29. Scuteri A, Orru M, Morrell CH et al. (2012) Associations of large artery structure and function with adiposity: effects of age, gender, and hypertension. The SardiNIA Study. *Atherosclerosis* 221, 189-197.
30. Johansen NB, Vistisen D, Brunner EJ et al. (2012) Determinants of aortic stiffness: 16-year follow-up of the Whitehall II study. *PLoS One* 7, e37165.
31. Wildman RP, Mackey RH, Bostom A et al. (2003) Measures of obesity are associated with vascular stiffness in young and older adults. *Hypertension* 42, 468-473.
32. Recio-Rodriguez JI, Gomez-Marcos MA, Patino-Alonso MC et al. (2012) Abdominal obesity vs general obesity for identifying arterial stiffness, subclinical atherosclerosis and wave reflection in healthy, diabetics and hypertensive. *BMC Cardiovasc Disord* 12, 3.
33. Brunner EJ, Shipley MJ, Ahmadi-Abhari S et al. (2015) Adiposity, obesity, and arterial aging: longitudinal study of aortic stiffness in the Whitehall II cohort. *Hypertension* 66, 294-300.
34. Marrugat J, Vila J, Baena-Diez JM et al. (2011) [Relative validity of the 10-year cardiovascular risk estimate in a population cohort of the REGICOR study]. *Rev Esp Cardiol* 64, 385-394.
35. Marti R, Parramon D, Garcia-Ortiz L et al. (2011) Improving interMediate risk management. MARK study. *BMC Cardiovasc Disord* 11, 61.

36. Satoh N, Shimatsu A, Kato Y et al. (2008) Evaluation of the cardio-ankle vascular index, a new indicator of arterial stiffness independent of blood pressure, in obesity and metabolic syndrome. *Hypertens Res* 31, 1921-1930.
37. Shirai K, Utino J, Otsuka K et al. (2006) A novel blood pressure-independent arterial wall stiffness parameter; cardio-ankle vascular index (CAVI). *J Atheroscler Thromb* 13, 101-107.